# Implicit HbA1c Achieving 87% Accuracy within 90 Days in Non-Invasive Fasting Blood Glucose Measurements Using Photoplethysmography

**DOI:** 10.3390/bioengineering10101207

**Published:** 2023-10-16

**Authors:** Justin Chu, Yao-Ting Chang, Shien-Kuei Liaw, Fu-Liang Yang

**Affiliations:** 1Department of Electronic and Computer Engineering, National Taiwan University of Science and Technology, No. 43, Sec. 4, Keelung Rd., Taipei City 10607, Taiwan; nk95061313@gmail.com (J.C.); skliaw@mail.ntust.edu.tw (S.-K.L.); 2Research Center for Applied Sciences, Academia Sinica, 128 Academia Rd., Sec. 2, Nankang, Taipei City 115-29, Taiwan; 3Division of Cardiology, Department of Internal Medicine, Taipei Tzu Chi Hospital, Buddhist Tzu Chi Medical Foundation, No. 289, Jianguo Rd., Xindian Dist., New Taipei City 231-42, Taiwan; necrosparkeps@tzuchi.com.tw

**Keywords:** photoplethysmography, HbA1c, blood glucose

## Abstract

To reduce the error induced by overfitting or underfitting in predicting non-invasive fasting blood glucose (NIBG) levels using photoplethysmography (PPG) data alone, we previously demonstrated that incorporating HbA1c led to a notable 10% improvement in NIBG prediction accuracy (the ratio in zone A of Clarke’s error grid). However, this enhancement came at the cost of requiring an additional HbA1c measurement, thus being unfriendly to users. In this study, the enhanced HbA1c NIBG deep learning model (blood glucose level predicted from PPG and HbA1c) was trained with 1494 measurements, and we replaced the HbA1c measurement (explicit HbA1c) with “implicit HbA1c” which is reversely derived from pretested PPG and finger-pricked blood glucose levels. The implicit HbA1c is then evaluated across intervals up to 90 days since the pretest, achieving an impressive 87% accuracy, while the remaining 13% falls near the CEG zone A boundary. The implicit HbA1c approach exhibits a remarkable 16% improvement over the explicit HbA1c method by covering personal correction items automatically. This improvement not only refines the accuracy of the model but also enhances the practicality of the previously proposed model that relied on an HbA1c input. The nonparametric Wilcoxon paired test conducted on the percentage error of implicit and explicit HbA1c prediction results reveals a substantial difference, with a *p*-value of 2.75 × 10^–7^.

## 1. Introduction

### 1.1. Non-Invasive Blood Glucose (NIBG) Measurements

Diabetes poses a great threat to global health care. The number of people with diabetes rose from 108 million in 1980 to 422 million in 2014 [1]. More than 1.4 million cases of newly diagnosed diabetes mellitus were documented in US adults aged 18 or older in 2019 alone, and diabetes was also responsible for more than 6.7 million deaths worldwide in 2021 [2,3]. Diabetes has a prevalence of about 1 in 10 adults and causes disability and severe end-organ damage, including renal failure, retinopathy, and nerve damage, if not well controlled. The introduction of insulin and anti-diabetic medications reduces the microvascular complications of diabetes in clinical trials but also raises the concern of hypoglycemia [4]. Thus, to achieve tight glycemic control, the use of easy, convenient, point-of-care devices to supervise glucose levels is pivotal in diabetes care.

Non-invasive blood glucose (NIBG) measurement refers to the process of determining blood sugar levels without the need for traditional methods that require pricking the skin to obtain a blood sample. Traditional finger-prick measurements can cause pain and carry a risk of infection, which might discourage individuals who need to monitor their blood glucose regularly. NIBG measurement offers a more comfortable and less intrusive way to monitor glucose levels.

Numerous methods based on diverse technologies have been explored for NIBG measurement. These include enzymatic methods that test saliva, tears, and body sweat [5,6,7], electromagnetic wave sensing methods that cover a wide area of the electromagnetic spectrum [8,9,10], and transdermal methods that measure the user’s bioimpedance [11].

Among these, photoplethysmography (PPG) stands out as a highly promising technology due to the fact that it is very easy to use and has very versatile applications [12]. PPG is a technology that monitors the light-absorption changes on the measured site. PPG sensors consist of a light-emitting diode (LED) that provides a stable light source, and light sensors that monitor the light intensity. While the blood volume at the measured site changes with pulsation, the light intensity also changes due to absorption and scattering. PPG allows unobtrusive continuous measurement while requiring only a single point of contact. Recent studies have even presented a breakthrough with contactless camera PPG for long-term, contactless, and continuous monitoring [13]. Various PPG-based applications are already being incorporated into commercially available products for the measurement of SpO2, stress levels, and blood pressure, and the detection of arrhythmias like atrial fibrillation.

Although the potential of relying on PPG alone to accurately estimate fasting blood glucose has frequently been explored via promising experimental results, a definitive answer remains elusive. We are of the opinion that the primary challenge stems from the missing variable or correction factor that addresses personal deviation. Every individual is inherently distinct, not only genetically but also in terms of their lifestyle and diet. Consequently, this gives rise to significant and undocumented variations among individuals when attempting to construct models for precise blood glucose level estimation.

Numerous NIBG studies based on PPG technology have been published over the past decades, employing a variety of different methods [14]. However, many of these studies suffer from small sample sizes and potentially compromise their generalizability. The most commonly employed PPG-extracted features are the morphological and heart rate variance features, which are correlated with an individual’s vascular function and autonomic neuropathy [15,16]. Many studies also explore features in different domains, utilizing techniques such as fast Fourier transform (FFT), Kaiser–Teager energy (KTE), and spectral entropy [17]. It is worth noting that using an excessive number of features can lead to overfitting, while using too few features may result in a lack of vital information required for accurate blood glucose level estimation. To date, the quest for an accurate yet simple standard for a medical NIBG meter remains unfulfilled.

### 1.2. From Measured HbA1c to Implicit HbA1c

In our previous studies, we showcased that a universal model incorporating quarterly HbA1c measurements as input features could substantially enhance model accuracy. However, we also observed that the presence of various medications had a detrimental impact on model performance. As a result, despite the incorporation of HbA1c, our ability to generate accurate estimations remains restricted for subjects not influenced by the effects of medication [18].

HbA1c, also known as glycated hemoglobin and sometimes referred to as hemoglobin A1C, is a crucial metric used to assess long-term blood glucose control and an important indicator for diagnosing diabetes [19]. It reflects the average blood glucose level over the preceding two to three months. Over time, the glucose in the bloodstream binds itself to the hemoglobin protein. The higher the blood glucose concentration, the more glucose is bound to the protein. While HbA1c is proven to be a strong feature that can significantly enhance prediction accuracy, it is not without shortcomings. Our previously proposed HbA1c model faces the challenge of a difficulty in taking HbA1c measurement. HbA1c measurements are not as easily acquired as a finger-prick blood glucose test. HbA1c measurements are generally only available in hospitals or specialized clinics. They require more specialized equipment compared to a conventional finger-prick BGL measuring device that in comparison is already commonly available for household use. The ability to acquire a simple alternative HbA1c could significantly improve the usability of prediction models that utilize HbA1c.

To address this challenge, we herein introduce an implicit HbA1c technology aimed to achieve accurate measurement of fasting blood glucose levels using photoplethysmography alone. In Figure 1 we present a schematic depiction of the concept for this work. When working with a model that only employs PPG-extracted features to estimate blood glucose levels, sparse prediction results often emerge due to overfitting or underfitting, symbolized by the large size of the circle. To refine the circle size of the predicted blood glucose levels using the PPG feature vector, we incorporate HbA1c into the model. This addition significantly reduces the overfitting or underfitting of the dispersed BGL prediction from PPG. HbA1c is the single most meaningful variable with a significant impact on model accuracy as it is correlated to the actual BGL. It is important to clarify our terminology: in this study, we refer to the measured actual HbA1c as “explicit HbA1c” since we use this measured value from the user’s result directly as it is. This is in contrast to our proposed “implicit HbA1c”. Implicit HbA1c is related to explicit HbA1C but is acquired through a novel use of the same HbA1c-enhanced model to measure BGL. Table 1 showcases the straightforward nature of the implicit HbA1c method by outlining the required information input from users to operate this model. Users only need to perform a single finger-prick measurement as a pretest to derive their implicit HbA1c, which encompasses systematic correction elements that account for personal deviations, thereby further enhancing the accuracy of blood glucose level predictions.

From the user’s perspective, the processes of determining implicit Hba1c are all carried out seamlessly behind the scenes. The users only need to take a single finger-prick BGL test at their initial pretest stage. The HbA1c values are all hidden from the user’s view, which is why the term “implicit HbA1c” is appropriate.

## 2. Experiments and Method

### 2.1. Experimental Set-Up

From the original dataset comprising 2632 entries, a subset of 856 entries of data consisting of data from subjects not undergoing drug treatment was meticulously chosen for this study. The dataset is collected from twenty local healthcare centers across Taipei and Taoyuan County with random voluntary participants. During the data collection phase, most of the lower blood glucose subjects were unwilling to participate in the second testing, thus their data were used in model training exclusively. On the other hand, higher blood glucose subjects displayed greater enthusiasm for further testing after a few weeks to monitor changes in their blood glucose levels, as shown in Table 2. Each entry within this subset comprises two consecutive 60 s segments of PPG measurement at a 250 HZ sampling rate collected through transmissive PPG finger clips (infrared, wavelength of 940 nm) on the index finger with the TI AFE4490 Integrated Analog Front End, along with corresponding measurements of blood glucose levels via finger-pricking using the Roche Accu-Chek mobile, HbA1c using the Siemens DCA Vantage Analyzer, and blood pressure using the Omron HEM-7320. The subjects were first asked to sit on the chair in a relaxed position for at least 5 min before the measurements started. During measurement, the blood pressure and finger-prick blood glucose level measurements were taken first, immediately followed by the 60 s long PPG measurement. The collection of these samples received approval from the Institutional Review Board of the Academia Sinica, Taiwan (Application No: AS-IRB01-16081). It is noteworthy that all subjects were comprehensively informed and consented to the collection of the data and their usage.

The 60 s long PPG signals are segmented into windows with a width of 400 data points (equivalent to 1.6 s) based on each pulse valley. A total of 11 features are extracted, encompassing both morphological and heart rate variance (HRV) features. The morphological features include the width of the pulse at half-height, the time taken from pulse valley to pulse peak, the sum of the pulse area of the minute, the average pulse area, and the median of the pulse area. The HRV features include the high, low, and total frequency power from fast Fourier transform (FFT), the percentage of pulse successive interval changes exceeding 20 ms, and the standard deviation of pulse successive interval changes.

In this study, our primary focus is exclusively on subjects who are not undergoing treatment with drugs. This approach serves as a follow-up to our previously proposed method, with the intent of enhancing its effectiveness. Our previous work achieved over 90% accuracy on cohorts not affected by medication with measured HbA1c employed as a feature.

For this work, subjects with multiple entries are deliberately reserved for use as the testing set, while the remaining subjects constitute the training set. The characteristics of both the training and testing sets are concisely outlined in Table 2. To align our approach with practical usage scenarios, a total of 61 pairs, each with a time interval not exceeding 90 days, are utilized for testing. Evaluating performance beyond the three-month validity of HbA1c would be both impractical and devoid of meaningful insights. Thus, the data pairs with intervals exceeding 90 days were further excluded from the testing. Within the testing set, each subject’s multiple rounds of measurements are meticulously paired together in a sequential manner to establish the testing data structure. Each pair is composed of a pretest and a test measurement, collected from different measurement rounds, thereby forming varying time intervals. The valid time interval between the pretest and test spans from 11 to 90 days. Notably, none of the measurements belonging to subjects designated for the testing set are used in the training set. This meticulous separation ensures the establishment of the strictest testing condition, where the model has not yet been influenced by any prior measurements of the intended test subjects.

### 2.2. Method

The methodology proposed in this study revolves around the utilization of a pretest round to derive an implicit HbA1c value, subsequently enhancing the accuracy of blood glucose level (BGL) predictions during the testing round.

The workflow of this method is depicted in Figure 2. Firstly, a set of three BGL prediction models are trained employing PPG signals, PPG-extracted features, and HbA1c as inputs. These models use identical structures and will be the only prediction models used throughout the method. The three models’ purpose is to validate outcomes by cross-referencing with each other. During the pretest phase, a PPG measurement with a corresponding finger-prick BGL reading is taken and inversely applied to the prediction model. The pretest PPG data are then input into the model with varying HbA1c values ranging from 4 to 12 in increments of 0.1. Consequently, a series of BGL predictions is generated, as exemplified in Figure 3. Among these predictions, the value closest to the measured finger-prick BGL reading is identified, and the corresponding HbA1c value is selected as the implicit HbA1c. This process is repeated independently for each of the three models. The disparities among the outcomes from the three models are assessed to ensure they fall within an acceptable threshold for consistency. Among the three results, the median value is chosen to serve as the designated implicit HbA1c.

During the testing phase, only the PPG measurement is collected and then joined with the pretest-determined implicit HbA1c as input for the model to generate the BGL prediction. Once more, both the PPG measurement and implicit HbA1c are independently input into the three models, and the differences among the prediction outcomes are assessed to ensure their consistency. In the event that the differences among models with identical structures and the same input data exceed a certain threshold, the results are deemed unreliable and should be disregarded. This process represents a straightforward and simple approach, leveraging preexisting models to obtain an alternative HbA1c value that enhances the accuracy of BGL predictions.

In this study, we utilized Python version 3.9.13 and Keras version 2.7 with tensorFlow version 2.7 as the backend for model building. The BGL prediction model utilized in this study used an identical structure to our prior HbA1c-based method, thus facilitating objective comparisons. The detailed model structure design with every layer can be found in Appendix A. The model comprises two parallel one-dimensional convolutional neural network (CNN) blocks, each featuring different filter lengths to encompass both micro and macro perspectives of the input signal window. The CNN outputs are subsequently concatenated with a manually extracted feature vector, which includes the HbA1c measurement. This combined information is then passed through several fully connected layers to generate the BGL prediction output. A comprehensive depiction and in-depth design of the model’s structure can be found in our earlier work, where the model achieved a prediction accuracy of over 94% within Clarke’s error grid (CEG) zone A for subjects not influenced by any form of medication.

## 3. Results

In this study, Clarke’s error grid (CEG) analysis is used as the main performance indicator, as the ISO 15197:2013 (International Organization for Standardization) recommendation requires personal use glucose meters to have 99% of the measurement within CEG’s zones A and B [20]. Clarke’s error grid analysis is a graphical method used to evaluate the accuracy of blood glucose meters developed by David Clarke in 1987 as a way to assess the clinical significance of errors in glucose measurements. CEG consists of five zones from A to E, each reflecting different clinical significance [21]. Zone A represents an accurate prediction where any differences between the prediction and reference values are considered negligible. Zone B reflects a prediction with a clinically acceptable error which could lead to unnecessary treatment but does not have a significant impact. As for Zones C to E, they represent different degrees of danger to users; if the result is used for clinical purposes, it could lead to severe harm or even death.

In Figure 4a, we presented the difference between implicit HbA1c and its corresponding measured reference HbA1c. The graph exhibits a rough alignment with the diagonal line, while implicit HbA1c values appear systematically higher than their corresponding explicit HbA1c values. Implicit HbA1c reflects the HbA1c value that the model anticipates given a specific blood glucose level. This phenomenon can be attributed to the fundamental difference between the training and testing sets we used. As a result, the training set we used (subjects without multiple entries of measurement) is predominantly composed of subjects with lower blood glucose level. In contrast, the testing set is predominantly composed of individuals with prediabetes and diabetes. Due to the methodology employed, the testing set required the test subjects to have two measurements (pretest and test), but the training set did not require a pretest for model training. This makes it impossible to mix the data between training and testing sets to achieve a more balanced distribution between the two sets. Table 2 provides a glimpse of the notable differences in average HbA1c and blood glucose levels between the training and testing datasets. From Figure 4b we can see the clear difference in the distribution of the BG–HbA1c relation between our training and testing datasets. Consequently, as we proceed with the process of calculating implicit HbA1c, the elevated fasting blood glucose levels observed within the testing subjects contribute to higher implicit HbA1c values. The systematic deviation between the explicit and implicit HbA1c values does not reflect the error; instead, it shows the amount of correction items adjusted by the methodology to bridge the population for an accurate BGL estimation.

For comparison, a set of predictions was also conducted using explicit HbA1c. This comparative analysis was carried out using the same testing dataset. The overall prediction performances by CEG’s zone ratios are documented and summarized in Figure 5 shown below. In the figure, we can see the difference in prediction ability between using the newly proposed implicit HbA1c and the previously used explicit HbA1c. Overall, while using implicit HbA1c, the model not only alleviates the inconvenience associated with HbA1c measurement, but also leads to a substantial 16% improvement in prediction accuracy. This outcome is an indication that implicit HbA1c can be more effective than measured HbA1c. This intriguing phenomenon is hypothesized to stem from the implicit HbA1c calculation process which also introduced a degree of adjustment of personal deviation.

The distribution of the prediction percentage error of using the implicit HbA1c and explicit HbA1c methods is presented in Figure 6. In the figure, we can see the prediction percentage errors from the implicit HbA1c method exhibit a normal distribution, while those from the explicit HbA1c method displayed a left-skewed distribution with systemically lower prediction. To ascertain the significance of the difference in model performance, a nonparametric Wilcoxon signed-rank test was conducted based on the percentage errors. The statistical result revealed that there are significant differences in prediction accuracy between the two methods, with a *p*-value of 2.75 × 10^–7^, much smaller than the significance level of 0.05.

## 4. Discussion

The accurate estimation of blood glucose levels (BGL) from non-medicated subjects can be achieved through a machine learning (ML) model that utilizes both photoplethysmography and HbA1c input, as we have previously demonstrated in our work published in *Sensors*. In that study, the HbA1c measurements used were taken simultaneously under the assumption that they could represent any recently measured HbA1c value with limited degradation in performance, given that HbA1c reflects a three-month average of blood glucose concentration. The less-than-ideal performance on the prediction results when using explicit HbA1c in this study was expected due to the previously mentioned disparities between the training and testing sets, as well as the increased time interval when compared to our prior work. Despite these increased challenges, the implicit HbA1c method effectively generates accurate predictions. This highlights the efficacy of implicit HbA1c in covering correction items from personal deviations.

A machine learning model for BGL estimation can generally be represented as Equation (1). Here, the function ML() symbolizes the machine learning model, while *F*_1_ through *F*_n_ correspond to the diverse set of features that collectively contribute to achieving an accurate prediction of the blood glucose level.
BGL = ML(*F*_1_, *F*_2_, *F*_3_ … *F*_n_)(1)

While different methods may employ different features, our prior work demonstrated that BGL can be accurately estimated by a machine learning model with PPG and HbA1c input, albeit under certain conditions. This leads us to modify Equation (1) into Equation (2a):BGL = ML(*PPG*, *HbA1c*)(2a)

However, it is important to acknowledge the intricate interplay of variables such as medication, individual differences, lifestyle variations, and more, which may not have been fully accounted for. This realization prompts us to introduce the correction item ∑C_i_ into the equation. For subjects not undergoing treatment with drugs, the effects of ∑C_i_ may not be significant enough to seriously hinder the prediction performance, but it is undeniable that these effects still exist. Consequently, we further revise the equation into Equation (2b).
BGL = ML(*PPG*, *HbA1c*, ∑C_i_)(2b)

These personal difference effects were dealt with by using a personalized deduction learning model that required multiple measurements from the user in our previous work [22]. Other works sought to account for these deviations by utilizing numerous personal profiles [23]. In this study, we leverage the concept of implicit HbA1c to achieve a similar effect.

Implicit HbA1c is determined by substituting HbA1c and BLG in Equation (2). It is like solving a multi-variate polynomial function with only one unknown variable. To solve for the unknown HbA1c value, the model is provided with a range of HbA1c inputs, generating a series of predictions. By cross-reference these predictions with the known BGL value, we can determine which corresponding HbA1c produces the most accurate estimation. This process not only yielded an HbA1c estimate, but it also accounted for the aforementioned correction items ∑C_i_. In other words, implicit HbA1c is the HbA1c value that has been adjusted to accommodate an individual’s specific correction items. Thus, this refinement further transforms Equation (2b) into Equation (3)
BGL = ML(*PPG*, *HbA1c_imp_*)(3)

HbA1c reflects an average BGL, and its correlation with fasting BGL is influenced by individual lifestyle, such as constant high BGL during the day and multiple meals. Consequently, the relationship between each individual’s HbA1c and fasting blood glucose follows a unique curve. For instance, individuals with prediabetes may still have a pancreas capable of producing a sufficient amount of insulin to maintain normal fasting BGL, but their daily BGL may fluctuate in a big range depending on diet and lifestyle. We anticipation that the proposed method will demonstrate effectiveness across various demographics, including different races, ages, and genders, as it effectively compensates for personal deviations arising from miscellaneous correction factors.

The self-monitoring of blood glucose (SMBG) serves as an indicator of daily sugar control status in modern diabetes treatment, and its importance might be introducing behavior changes, improving glycemic control, and optimizing therapy [24]. Intensive insulin therapy is usually accompanied by daily SMBG and has proved to reduce the end-organ damage in patients with insulin-dependent diabetes mellitus [25]. There is also evidence suggesting the benefit in pre-diabetic patients or those under oral anti-diabetic drugs [26]. Some diabetes guidelines suggest SMBG use not only while fasting but also in the post-prandial stage, because the post-prandial glucose excursion, measured by the delta change in fasting and post-prandial sugar, has been demonstrated to correlate with cardiovascular risk [24]. Hence, the structured SMBG protocol by performing glucose tests before and after a meal in pairs has been evaluated in clinical trials and improves glycemic control [27]. Our implicit HbA1c method may increase the frequency of sugar monitoring compared to the guideline-suggested 2~3 times of SMBG per week in non-insulin-treated T2DM; the usage of this novel non-invasive glucose monitor technology might help diabetologists to optimize diabetic therapy in the future. However, our original dataset was collected in a fasting population; thus, the reliability of post-prandial sugar use remains uncertain. In addition, our prediction model in the insulin-treated population, whose SMBG assessments are in most demand, is less powerful than those not undergoing drug treatment. A further improvement of our algorithm and studies including a broader spectrum of diabetic populations are mandatory.

## 5. Conclusions

The significance of HbA1c as a valuable feature for non-invasive blood glucose prediction is widely acknowledged, although the inconvenience of acquiring HbA1c measurements remains. The HbA1c measurements are generally only available in hospitals or specialized clinics. To tackle this issue, this study introduced an innovative approach known as implicit HbA1c value which derives an alternative HbA1c that only requires a single finger-prick blood glucose measurement and can be easily conducted at home by the users. Implicit HbA1c was introduced as a solution to enable accurate glucose predictions without the need for direct HbA1c measurements with specialized equipment, and it also demonstrated the ability to further improve the prediction performance. The implicit HbA1c method achieved 87% of the prediction results within CEG’s zone A, and the remaining 13% close to the zone A boundary. The implicit HbA1c approach not only exhibited a remarkable 16% improvement over the measured HbA1c method by covering personal correction items automatically, but also demonstrated an extended prediction validity period with testing data from 11 up to 90 days. The nonparametric Wilcoxon paired test conducted on the percentage error suggests a statistically significant difference between their performances with a *p*-value of 2.75 × 10^−7^.

## Figures and Tables

**Figure 1 bioengineering-10-01207-f001:**
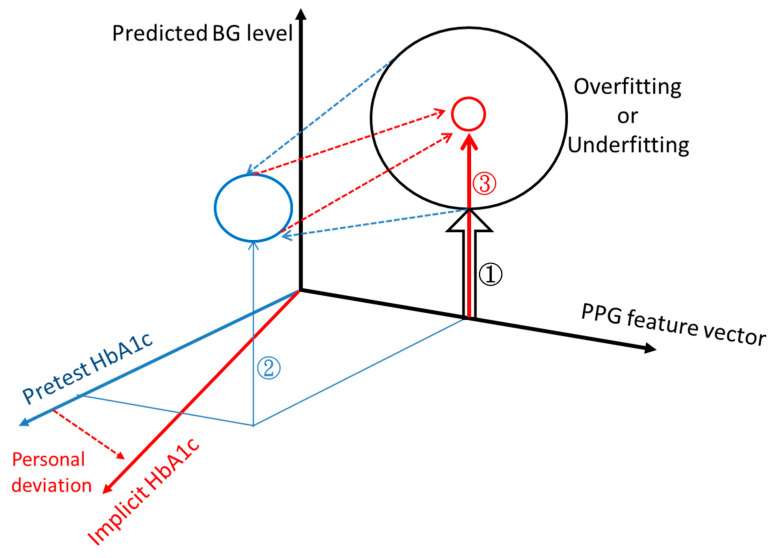
Schematic illustration of the concept of overfitting or underfitting using circle size, along with the transition from explicit HbA1c to implicit HbA1c for an improved machine learning prediction of blood glucose levels. To reduce the circle size of the predicted blood glucose level with the PPG feature vector, we introduce pretest HbA1c. It significantly reduces instances of overfitting or underfitting that lead to scattered blood glucose level predictions derived solely from PPG data (from ① to ②). Implicit HbA1c is related to pretest HbA1c but contains systematic correction items to account for personal deviation, thereby further enhancing the accuracy of blood glucose level prediction (from ② to ③). The input features for each stage are as follows: ① PPG features, ② PPG features and pretest HbA1c, ③ PPG features and implicit HbA1c.

**Figure 2 bioengineering-10-01207-f002:**
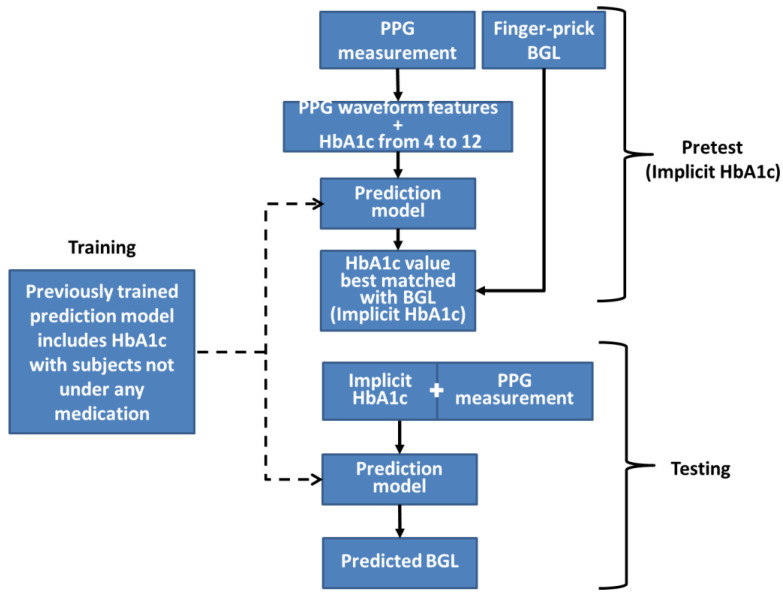
Illustration of the sequential workflow, starting from the pretest phase to derive the implicit HbA1c, followed by its application during testing to generate the final blood glucose level prediction. During pretest, the required inputs are PPG measurement and a finger-prick blood glucose level. The finger-prick value is only used at the final step of the pretest to determine implicit HbA1c value. During testing, the required inputs are the pretest determined implicit HbA1c and a PPG measurement.

**Figure 3 bioengineering-10-01207-f003:**
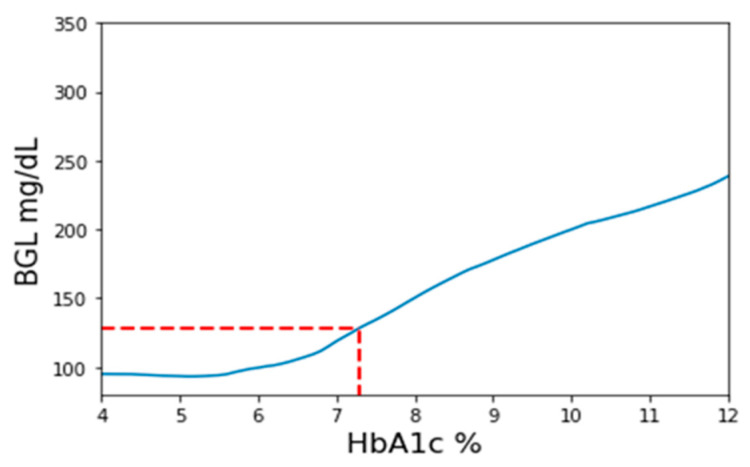
Illustrative example of the process for determining implicit HbA1c. A range of HbA1c values, ranging from 4 to 12, is employed to estimate blood glucose levels (BGL). Among these values, the HbA1c value corresponding to the predicted BGL closest to the pretest reference BGL is identified and designated as the implicit HbA1c value.

**Figure 4 bioengineering-10-01207-f004:**
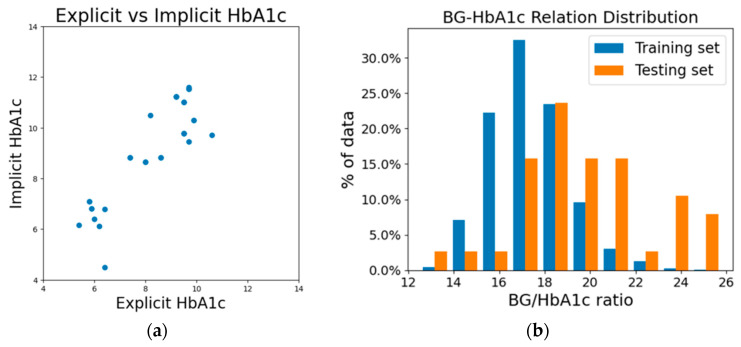
(**a**) Difference in implicit HbA1c and corresponding measured reference HbA1c, and (**b**) distribution of the BG–HbA1c (measured) relationship for the training and testing sets.

**Figure 5 bioengineering-10-01207-f005:**
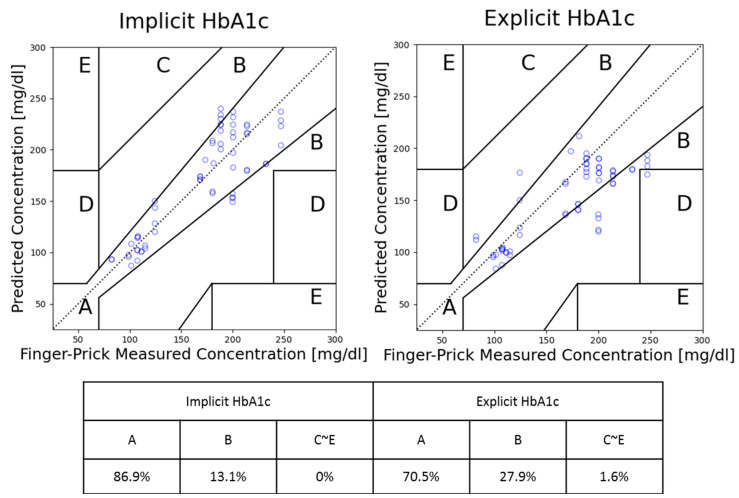
CEG analysis of using implicit HbA1c and explicit HbA1c for model prediction on subjects not affected by drugs.

**Figure 6 bioengineering-10-01207-f006:**
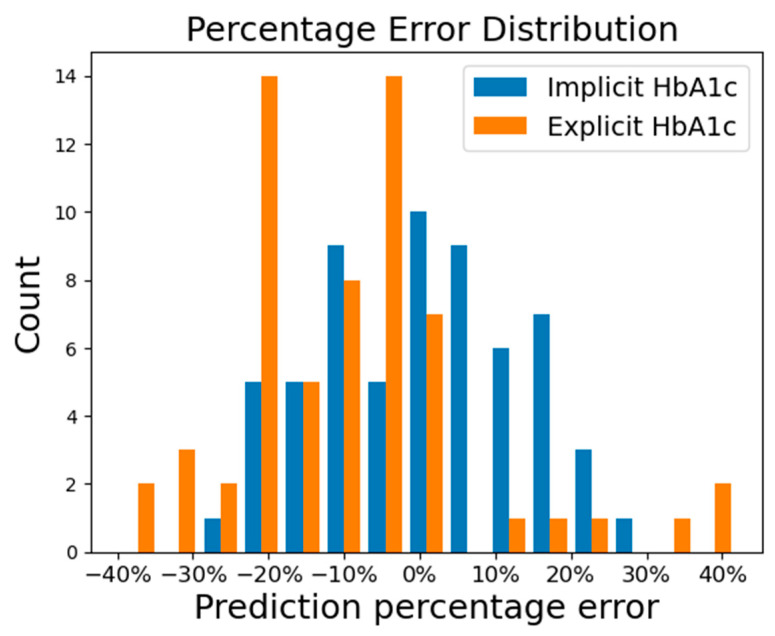
Percentage error distribution of the prediction while using implicit and explicit HbA1c for conducting the nonparametric Wilcoxon signed-rank test. The implicit HbA1c method demonstrates a distribution pattern that resembles a normal distribution. The explicit HbA1c method exhibits a left-skewed distribution with systemically lowered prediction.

**Table 1 bioengineering-10-01207-t001:** Summary of the information input and output for each section of the process for the proposed implicit HbA1c method.

	**Model training**	**User**		
		**Featuring HbA1c** **(Pretest)**	**Interval up to 90 days**	**Testing**
**Required Input**	PPG signal Reference HbA1c Reference BGL	PPG signal Reference BGL		PPG signal Implicit HbA1c
**Outcome**	BGL prediction model	Implicit HbA1c		Predicted BGL

**Table 2 bioengineering-10-01207-t002:** Characteristics of the subjects in the training and testing sets with their mean ± standard deviation.

	Dataset	Interval between Test and Pretest	BG (mg/dL)	HbA1c (%)	Age (Years)	BMI (kg/m^2^)
Total 856 entries	Training (747 entries)	No pretest	99.9 ± 12.9	5.7 ± 0.53	57.9 ± 9.7	23.4 ± 3.2
	Testing (61 pairs)	45 ± 19 days	154.9 ± 50.8	7.7 ± 1.76	62.7 ± 3.95	28.4 ± 4.3

## Data Availability

Data sharing is not applicable to this article.

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
