# Peer review of "Implicit HbA1c Achieving 87% Accuracy within 90 Days in Non-Invasive Fasting Blood Glucose Measurements Using Photoplethysmography"

_bioengineering, 2023, doi:10.3390/bioengineering10101207_

Round 1
Reviewer 1 Report
In current manuscript, the authors presented a methodology to predict fasting blood glucose measurement with an added implicit HbA1C parameter. Although the approach of using PPG is technically sound, the necessity of finger prick pretest for the prediction model is still inconvenient and invasive. With only an incremental improvement in accuracy over existing methods, the authors must further justify the significance of this study and consider to include additional parameters to improve the model.
N/A
Reviewer 2 Report
Authors have proposed a novel method to take into account the concentration of glycated hemoglobin (HbA1c) from an implicit ML model, i.e. using only the results of the standard results of photoplethysmography analysis. The resulting model allows for better prediction of non-invasive fasting blood glucose levels for diabetes monitoring, including not only the concentration of HbA1c, but also personal characteristics of the patient. Overall, the improved model may find applications in diabetes diagnostics. A few additional information can help to remove questions of the interested reader (minor revision).
- It is not quite clear how the model takes into account the implicit HbA1c level in the test set. Authors say that predicted HbA1c level is used along with PPG results, but as a separate piece of data. For the train set, the implicit HbA1c level was predicted based on the comparison of prediction result with the experimental data. Do authors use a NN or a analytical function with numerical parameters for implicit HbA1c level?
- Authors say that they used a larger number of healthy patients in the train set than in the test set, which led to a systematic error. If the HbA1c level was measured for the patients from the test set, then it is not clear why the results of the two sets cannot be mixed to make equal numbers of healthy and unhealthy patients in both sets? This issue requires either an explanation, or the additional tests with mixed datasets.
Reviewer 3 Report
Review of the manuscript entitled: Implicit HbA1c Achieving 87% accuracy for Non-invasive Fasting Blood Glucose Measurements within 90 days by Photoplethysmography.
Non-invasive monitoring of blood glucose levels is currently a very hot topic. Many new solutions are regularly proposed on this issue. The authors successfully continue the research on glucose measurements begun in their previous works [18, 22]. I believe that the manuscript deserves publication after a small addition to the text, which will help a wide range of readers fully understand the proposed method.
1. The authors should explain in a few words the basic physical principles of photoplethysmography (PPG). More importantly, they must describe their measurement methodology: what type of LED, wavelength, light absorption measurement, what part of the human body is being examined.
2. The authors should add information about the type, manufacturer, and licensing of the conventional finger prick BGL measuring device.
3. I ask the authors to add details of the machine learning model (or one-dimensional convolutional neural network) and the codes used.
4. I don't really like the title of the article. What does “Implicit HbA1c” mean? The authors introduce a new scientific term in the title, but explain its meaning only on page 3. I'm not sure this term will be accepted by the scientific subcommunity.
A small request.
Page 2:
… commercially available products namely SPO2, stress…
Please write SpO2.
In conclusion, I wish the authors the implementation of their research into clinical practice.
